# Supercritical CO_2_ Impregnation of Thymol in Thermoplastic Starch-Based Blends: Chemico-Physical Properties and Release Kinetics

**DOI:** 10.3390/polym14204360

**Published:** 2022-10-16

**Authors:** Marija Lucic Skoric, Stoja Milovanovic, Irena Zizovic, Rodrigo Ortega-Toro, Gabriella Santagata, Mario Malinconico, Melina Kalagasidis Krusic

**Affiliations:** 1Innovation Center of the Faculty of Technology and Metallurgy, University of Belgrade, Karnegijeva 4, 11120 Belgrade, Serbia; 2Faculty of Technology and Metallurgy, University of Belgrade, Karnegijeva 4, 11120 Belgrade, Serbia; 3Faculty of Chemistry, Wroclaw University of Science and Technology, Wybrzeze Wyspianskiego 27, 50-370 Wroclaw, Poland; 4Food Packaging and Shelf Life Research Group (FP&SL), Food Engineering Program, Universidad de Cartagena, Avenida del Consulado Calle 30 No. 48-152, Cartagena de Indias 130015, Colombia; 5CNR, Institute for Polymers, Composites and Biomaterials, Via Campi Flegrei, 34, Pozzuoli, 80078 Napoli, Italy

**Keywords:** thermoplastic starch, PCL, thymol, supercritical CO_2_, impregnation

## Abstract

The aim of the present study was to investigate starch-based materials, prepared in an environmentally friendly way and from renewable resources, suitable for the development of biodegradable active food packaging. For this purpose, a bioactive compound (thymol) was incorporated into thermoplastic starch (TPS) and a TPS blend with poly (ε-caprolactone) (TPS–PCL) by the supercritical CO_2_ (scCO_2_) impregnation process. Impregnation experiments with scCO_2_ were carried out at a pressure of 30 MPa and temperatures in the range of 40–100 °C during 1 to 20 h. The structural, morphological, and thermal properties of the obtained materials were comprehensively evaluated. Bioactive component release kinetic studies were performed in water at 6 °C and 25 °C. It was shown that the scCO_2_ impregnation process could be successfully employed for thymol loading into TPS and TPS–PCL. The process was significantly influenced by the operating temperature and time as well as content of PCL. The samples showed a controlled release of thymol within seven days with a higher amount of released thymol from the TPS–PCL blend. The obtained materials are solvent-free and release the bioactive component in a controlled manner.

## 1. Introduction

The development of novel and biodegradable materials from renewable sources for different applications is the focus of scientific attention, primarily due to the raised awareness on the environmental impact of petroleum-based plastics, especially in the context of water, soil, and air pollution, as well as the increased omnipresence of microplastics [1,2,3]. It was shown that one of the promising alternatives to petroleum-based packaging plastics is starch due to its biodegradability, availability, renewability, and low cost [3,4,5]. Starch-based materials such as thermoplastic starch (TPS), with or without the addition of other biodegradable polymers, can be processed employing conventional equipment for the production of traditional thermoplastic materials and transformed by injection or blow molding into the final products suitable for food packaging [5,6,7,8]. For instance, our previous study reported that thermoplastic starch–poly (ε-caprolactone) (TPS–PCL) blends in the form of a film have gas and water vapor barrier properties comparable to synthetic plastic commonly used for food packaging [6]. In addition, it was reported that TPS biodegrades ca. 20 days after burial in soil, while TPS–PCL blends biodegrade during the period of ca. 20 to 150 days depending on the PCL content [9].

Beside biodegradable food packaging, active-packaging materials are one of the current trends in food technology [8]. This relates to packages containing additives that maintain or extend food quality or shelf-life and therefore contribute to the reduction in spoilage, food waste, food recalls, and foodborne illness outbreaks [10]. There are several reports on the preparation of TPS-based materials with bioactive components by conventional methods for the development of active food packaging [4,7,8,11,12]. These reports include the addition of watermelon rind extract to a water solution at 70 °C before solvent casting [4], the incorporation of TiO_2_ nanoparticles by extrusion at 165 °C [7], and the addition of copper particles by thermocompression at 140 °C [12]. However, one of the methods that proved to be more efficient for the development of advanced materials is supercritical solvent impregnation (SSI). The application of supercritical fluids in the encapsulation of active compounds is of great relevance to the food, pharmaceutical, and cosmetic industry [13]. SSI offers numerous and unique advantages over conventional processes overcoming their drawbacks, such as the use of organic solvents, high temperatures, low diffusion coefficients, long contact times, high consumption of additives, non-uniform distribution of impregnates, etc. [14]. This process allows for the incorporation of an active compound throughout the whole volume of solids in an environmentally friendly way [14,15]. The impregnation of an active compound in the carrier polymer occurs as a consequence of the interactions between an active compound, polymer, and supercritical fluid. This results in the adsorption or a physicochemical attachment of active compound molecules to the polymeric matrix [16]. The solvent power of the supercritical fluid can be easily changed, e.g., decreased with a pressure decrease during decompression, causing the precipitation of an active compound dissolved in the supercritical fluid within the polymer material [15,17]. Supercritical carbon dioxide (scCO_2_) is particularly suitable for impregnation processes due to its favorable critical parameters, high diffusivity in organic matter, the absence of surface tension, and ease of recovery [18]. The relevance of the SSI processes in the production of systems for the controlled release of active substances has considerably increased in the last decade [15,18,19].

Several studies justified the application of SSI for the incorporation of active compounds into starch-based materials, including the impregnation of *n*-octenyl succinate modified starch with lavandin essential oil [15], the impregnation of starch xerogels and aerogels with thymol [17], the impregnation of starch biocomposite films with cinnamaldehyde [1], the impregnation of nanoporous starch aerogels with phytosterol [20], and the impregnation of starch aerogel microspheres with ketoprofen and benzoic acid [21]. To the best of our knowledge, this is the first comprehensive study on the application of the SSI process for incorporation of thymol into biodegradable TPS and TPS–PCL blends. Thymol, a substance with a GRAS status (generally recognized as safe), is known for its strong anti-septic, antioxidant, and antimicrobial properties [18,22]. It was previously employed for the development of active food packaging [23]. Due to its high solubility in supercritical fluids, scCO_2_ is considered an effective medium for thymol impregnation into a polymer [24]. This study reports for the first time the influence of the PCL addition to TPS and the SSI process conditions (temperature and time) on the starch-based material properties, including controlled release of the bioactive component. The applied SSI process and the attained TPS-based materials showed potential to be applied in the development of biodegradable active food packaging in a sustainable and environmentally friendly manner.

## 2. Materials and Methods

### 2.1. Materials

For the preparation of thermoplastic starch without and with poly (ε-caprolactone) (TPS and TPS–PCL, respectively), corn starch, with a 25% amylose content according to the supplier data and purchased from Roquette (Roquette Laisa, Beni-faió, Spain), glycerol obtained from Panreac Química, S.A. (Castellar del Vallès, Barcelona, Spain), poly (ε-caprolactone) (M_n_ = 80.000 Da) provided by Fluka (Sigma-Aldrich Chemie, Steinheim, Germany), glycidyl methacrylate (purity 97%), and benzoyl peroxide (BPO) supplied by Aldrich Chemicals were used. Thymol (purity > 99%) used for the impregnation process was supplied by Sigma-Aldrich Chemie GmbH (Germany). Carbon dioxide (purity 99%) was supplied by Messer-Tehnogas (Belgrade, Serbia).

### 2.2. Methods

#### 2.2.1. Preparation of Thermoplastic Starch-Based Pellets

Samples of TPS and TPS–PCL were obtained by the extrusion process previously described in detail [6]. Briefly, the thermo-plasticization process was performed by blending native corn starch (100 g) with glycerol (30 g) and water (50 g) in a starch:glycerol:water weight ratio of 1:0.3:0.5 wt./wt. Then, the blend was extruded to obtain about 70 g of plasticized starch pellets (TPS, each pellet weighing about 30 mg and sizing up about 3 mm^3^).

Prior to the preparation of TPS–PCL samples, PCL was chemically modified by means of a radical grafting reaction with glycidyl methacrylate (PCL_G_), to obtain a compatibilizer agent of TPS–PCL blends. The reaction was performed in the melt, using a Brabender Plastograph EC GmbH & Co. KG batch mixer (Duisburg, Germany), equipped with two counter-rotating roller blades. To formulate the TPS–PCL blend, first 0.2 g of PCL was used for 1 g of TPS. Additionally, 5 g of PCL_G_ was added per 100 g of plain polymeric TPS–PCL blend to obtain the final formulation of the blend. Blends were obtained by melt processing using a co-rotating twin-screw extruder (Teach-Line^®^ZK 25T, Collin, Germany). The final blend strands were pelletized, thus obtaining pellets of 30 mg of average weight and 3 mm^3^ in mean size.

#### 2.2.2. Impregnation of Polymers with Thymol

Selected polymers (TPS and TPS–PCL) were impregnated with thymol using scCO_2_ in a high-pressure unit equipped with a view cell (25 mL, Eurotechnica GmbH, Bargteheide, Germany) previously described in detail [24]. Polymers (1 g) were placed in a porous basket above the thymol (3 g). Polymers and thymol were separated by a Teflon fabric to avoid possible splashing of the polymer with liquefied thymol during the process. The samples were exposed to CO_2_ under the desired pressure and temperature conditions for a time ranging from 1 h to 20 h. Venting out of the CO_2_ from the vessel followed at a rate of 5 MPa/min. The impregnation process was performed at a pressure of 30 MPa and temperatures of 40 °C, 70 °C, and 100 °C. Conditions were selected based on our previous reports on the impregnation of thymol [24,25,26] as well as preliminary tests which showed that the SSI kinetics were considerably slower at pressures lower than 30 MPa for tested polymers in comparison to the kinetics at 30 MPa. The impregnation experiments were performed in triplicate. A list of all prepared samples is given in Table 1.

Thymol loading (*L*, %) was calculated as the mass ratio of loaded thymol and the impregnated sample multiplied by 100%. The mass of loaded thymol was quantified using an analytical scale (accuracy ±0.01 mg) as the mass difference of the polymer before and after the impregnation. The mass of loaded thymol was confirmed by release tests in ethanol using a UV/VIS spectrophotometer (Shimadzu 1800, Kyoto, Japan).

### 2.3. Characterization of TPS-Based Samples

#### 2.3.1. Attenuated Total Reflection Fourier Transform Infrared (FTIR-ATR) Spectroscopy

Fourier transform infrared (FTIR) spectra of the selected samples were recorded using an ATR-FTIR spectrometer Nicolet iS10 (Thermo Fisher Scientific Inc., Madison, WI, USA) in the mid-IR region (500 cm^−1^–4000 cm^−1^). To evaluate the potential interactions of scCO_2_ and thymol with polymers, FTIR spectra of the non-treated polymers and thymol were also recorded.

#### 2.3.2. Thermal Analysis

Differential scanning calorimetry (DSC) measurements were performed by a TA DSC-Q2000 instrument equipped with a TA Instruments DSC cooling system under a nitrogen purge gas flow of 30 mL min^−1^. Indium was used to calibrate the calorimeter. Approximately 5 mg of the samples was placed into aluminum pans, sealed, and analyzed. All the specimens were equilibrated at 25 °C and heated up to 110 °C at 20 °C/min. Then, an isotherm step was performed at 110 °C for 60 min to remove the free and bound water. Later, the samples were cooled to −80 °C at 20 °C/min, thermally stabilized for 2 min at that temperature, and reheated up to 200 °C at 20 °C/min. For each sample, data analysis was averaged on a set of three measurements.

Thermogravimetric analysis (TGA) was performed by Mettler-Toledo TG-SDTA 851 thermobalance, under a nominal nitrogen flow of 30 mL min^−1^, in the temperature range of 25 °C–600 °C and at the heating rate of 20 °C min^−1^. The measurements were performed on samples of about 5 mg. For each sample, thermogravimetric tests were performed in triplicate.

#### 2.3.3. Scanning Electron Microscopy (SEM)

Morphological analysis of the samples was performed using a FEI Quanta 200 FEG Scanning Electron Microscope (SEM) on cryogenically fractured cross-sections. SEM observations were performed in low vacuum mode (PH_2_O: 0.7 Torr), using a large field detector (LFD) and an acceleration voltage of 5–20 kV. Prior to the observation, the sample surfaces were coated with a homogeneous layer (18 ± 0.2 nm) of Au–Pd alloy by means of a sputtering device (MED020, Bal-Tec AG).

#### 2.3.4. Mercury Intrusion Porosimetry

Mercury intrusion porosimetry (MIP) measurements of the selected samples were performed in the fully automated conventional porosimeter Carlo Erba 2000 series (pressure range: 0.1–200 MPa; pore diameter range: 7.5–15 000 nm) supplied with the Macropore 120 Unit and data processing acquisition software package Milestone 200.

### 2.4. Thymol Release Tests

Thymol release from the selected samples in water was tested in a static condition at temperatures of 6 and 25 °C (storage temperature of the food in cooling shelves and room temperature). Thymol concentration was determined at 274 nm using a UV/VIS spectrophotometer (Shimadzu 1800, Japan). Each thymol release test lasted for seven days, and measurements were performed in duplicate. Selected TPS and TPS–PCL samples impregnated with thymol, with a mass of 100 mg each, were immersed in 50 mL distilled water at 6 °C and 25 °C without stirring. At pre-determined periods, an aliquot (3 mL) of the released solution was removed and analyzed, and then returned to the release medium. Released thymol concentration was calculated using a previously determined calibration curve at 274 nm.

## 3. Results and Discussion

### 3.1. Impregnation of TPS and TPS–PCL with Thymol

Operating parameters of the SSI process such as pressure, temperature, and time are the key factors in the scCO_2_ impregnation process as they directly affect the dissolution of thymol in the scCO_2_ and the diffusion of the supercritical fluid mixture (scCO_2_ + thymol) into a solid carrier. The objective of the conducted experiments was the fabrication of materials with thymol in an environmentally friendly way. It was previously determined that the pressure of 30 MPa and temperatures of 40 °C, 70 °C, and 100 °C provide a high solubility of thymol in the scCO_2_ and a satisfactory loading of thymol into polymers [25,26]. Hence, these conditions were selected for the current research. The resulting kinetics of thymol loading into TPS are presented in Figure 1. As it can be seen, the loading of thymol increased with the impregnation time, as previously reported for other polymeric materials [22,27]. The exception was the process performed at 100 °C wherein the desorption of thymol was noticed after 6 h of impregnation. Additionally, thymol loading increased with an increase in scCO_2_ density from 662 kg/m^3^ (density of scCO_2_ at 30 MPa and 100 °C) to 910 kg/m^3^ (density of scCO_2_ at 30 MPa and 40 °C) except for the processes performed for 2 h wherein the loading was higher at 70 °C (density of scCO_2_ at 30 MPa and 70 °C is 788 g/m^3^) than at 40 °C. This is probably due to a higher diffusivity of scCO_2_ at higher temperature, which affected the process in the first hours more than the thymol solubility. Namely, due to an increase in density (lower temperature), the solubility of an active component in scCO_2_ increases as well [1,24]. Thus, thymol concentration in scCO_2_ is higher at 40 °C, leading to higher loading in later impregnation (longer than 2 h). The loading will increase until the maximal loading of the active component in the material is achieved [28] or until the affinity of the active component towards scCO_2_ prevails [15,18]. An increase in the impregnation time up to 20 h led to a decrease in thymol loading into TPS with the final values of 0.5, 0.7, and 0.1% at temperatures of 40, 70, and 100 °C, respectively. These relatively low thymol loadings are in agreement with the literature reports, which testify for starch’s low affinity towards bioactive substances during the scCO_2_ impregnation process. Sousa et al. [1] reported the maximal loading of cinnamaldehyde into starch films of 0.25% under the conditions of 25 MPa and 35 °C after 15 h of impregnation process, while Ubeyitogullari and Ciftci [20] reported the maximal loading of phytosterol into nanoporous starch aerogels of 0.99% at 45 MPa and 90 °C after 3 h. Additionally, Milovanovic et al. [17] reported thymol loadings into starch xerogels in the range of 0.6%–4.0% after 24 h of the SSI at 15.5 MPa and 35 °C. Similar results were obtained in this study for TPS (0.3%–2.9%, Figure 1) with a considerably shorter operating time (2–6 h).

Besides the impregnation of TPS with thymol, the impregnation of its blend with PCL was also evaluated. The addition of 20% of PCL to TPS and their blending made it possible to achieve considerably higher thymol loadings (Figure 2). While the highest amount of loaded thymol for TPS was around 3% (conditions: 30 MPa, 40 °C, 6 h), for the TPS–PCL blend it was around 15% (conditions: 30 MPa, 70 °C, 6 h). One of the reasons for the increase in thymol loading with an increase in PCL content in TPS from 0 to 20% could be the high affinity of thymol shown towards PCL during scCO_2_ impregnation processes. Namely, it was previously reported that thymol loading increases with an increase in the PCL content in PLA [29] and that thymol loading into neat PCL of 14.1% can be achieved after only 2 h at 30 MPa and 40 °C [25,30]. Under the same conditions, the thymol loadings into TPS and TPS–PCL obtained in this study were 0.6 and 3.7%, respectively. An additional reason for the increase in thymol loadings with PCL content in TPS could be the possible existence of voids between the phase interfaces due to low compatibility between TPS and PCL polymers. However, assessing the influence of the mentioned factors requires additional comprehensive investigation. Similarly to the results obtained for TPS, an increase in the impregnation time up to 20 h led to the decrease in thymol loading into TPS–PCL with the final values of 10.1%, 13.5%, and 8.9% at temperatures of 40 °C, 70 °C, and 100 °C, respectively.

The kinetics of thymol loading into TPS-PCL are significantly different compared to the kinetics of thymol loading into TPS. Namely, in the case of TPS–PCL, in the first four hours of the SSI, larger loadings were obtained at 70 and 100 °C compared to loadings at 40 °C, although the last state is characterized by the highest scCO_2_ density (910 kg/m^3^). The temperature of 100 °C (scCO_2_ density of 662 kg/m^3^) is the most favorable in the first two hours, while after four and six hours of the SSI, the largest loadings were obtained at 70 °C (scCO_2_ density of 788 kg/m^3^). The obtained results confirm that the interaction of the ternary system scCO_2_–polymer–thymol is complex and requires optimization for each system. Additionally, the obtained results show that the efficiency of thymol loading into TPS and TPS–PCL could be easily adjusted with a selection of the SSI process conditions.

The TPS and TPS–PCL samples selected for further analysis and characterization were the ones impregnated at 70 °C for 6 h (highest thymol load in the case of TPS–PCL) and 2 h (shorter impregnation time), as well as at 40 °C for 6 h (high loadings at low temperature).

### 3.2. Characterization of TPS and TPS–PCL Samples

#### 3.2.1. Attenuated Total Reflection Fourier Transform Infrared Spectroscopy on TPS-Based Materials

FTIR-ATR analysis was carried out to assess potential interactions between the thymol and TPS and the thymol and TPS–PCL after SSI at different processing conditions, and the obtained spectra are presented in Figure 3.

All TPS samples displayed a wide band in the range of 3600–3000 cm^−1^, which corresponds to the stretching vibration of the O–H groups of starch [31,32] (groups of amylose, amylopectin, and adsorbed water), glycerol, and thymol (Figure 3a). Additionally, a slight shift of O-H absorptions to 3200 cm^−1^ is observed in the samples containing thymol impregnated at longer times, independent of the temperature of the scCO_2_ process (6 h), and a higher percentage of impregnated thymol with a less intense band compared to neat TPS. This finding indicates that thymol has an effect on starch, which leads to a reduction in the interactions of hydroxyl groups in the starch matrix. Additionally, intermolecular hydrogen bonding occurs between thymol and starch, while thymol and glycerol (plasticizer) interact through –OH groups [33]. In the high-frequency region of spectra of neat and impregnated TPS, the bands originating from the asymmetric *ν*_as_ (CH_3_) and asymmetric *ν*_as_ (CH_2_) valence vibrations of C − H bonds appear around 2965 cm^−1^ and 2926 cm^−1^, respectively [31,34]. An additional band appears at 2866 cm^−1^ in the spectra of thymol-impregnated TPS samples, which is attributed to symmetric valence vibrations (*ν*_s_) of the C−H bonds in the −CH_3_ and −CH_2_ fragments. Another characteristic broad band occurring in the spectra of neat TPS around 1646 cm^−1^ is ascribed to the bending vibration of the O–H of bound water in the amorphous region of starch, and it could also be coupled with the asymmetric stretching vibrations of the O–H groups of glycerol (as plasticizer) [35]. This band disappears after the scCO_2_ impregnation process in all examined samples due to the hydrophobicity effect of thymol and the interaction between the functional group of thymol with starch as well as glycerol molecules that reduce the stretching vibrations of the O–H group [33]. In the impregnated samples, a new band at 1621 cm^−1^ is observed, which corresponds to aromatic ring C=C stretching vibrations of thymol [22]. Upon loading of thymol in TPS, in the region between 1650 and 1100 cm^−1^, different numbers and intensities of bands may be found (in comparison to the neat TPS spectrum), which may be associated with the presence of the benzene ring of thymol (1460–1420 cm^−1^). These bands could also indicate the physical interactions occurring between the starch and thymol. Furthermore, the band at 1359 cm^−1^ is assigned to the OH bending of the phenolic group [24]. Bands detected at 1285 cm^−1^ and 1250 cm^−1^ are assigned to a combination of OH deformation vibrations and C-O stretching vibrations [36]. Other adsorption bands in the lower region in all spectra are assigned to various functional groups, such as the band at 801 cm^−1^ attributed to out-of-plane aromatic C–H wagging vibrations [22].

FTIR-ATR spectra of neat and impregnated TPS–PCL are presented in Figure 3b. Besides all the above-mentioned bands attributed to the presence of starch, glycerol, and thymol, an additional important band in the region of 1760–1670 cm^−1^ is observed. It corresponds to the carbonyl stretching region of PCL. It is usually expected that, in the FTIR spectra of semicrystalline polyester, the carbonyl region is composed of two overlapping peaks: a relatively broad band centered at 1735 cm^−1^, occurring as a shoulder of a sharper and more intense band at 1720 cm^−1^, the first belonging to the PCL amorphous phase and the latter to the PCL crystals [37]. However, in the neat TPS–PCL spectrum an interesting change is noticed: the absence of a band corresponding to the amorphous phase and a shift of the crystalline band toward higher stretching vibration frequencies (1726 cm^−1^), suggesting that a new PCL crystalline domain was formed. However, in the impregnated TPS–PCL sample, the band corresponding to crystalline PCL is not observed, and the spectrum only shows two bands located at about 1734 cm^−1^ and 1708 cm^−1^ attributed, respectively, to free and hydrogen-bonded C꞊O groups in PCL [38].

#### 3.2.2. Surface Texture and Morphology Studies on TPS-Based Materials

In order to describe and better understand the observed changes of the neat and impregnated TPS and TPS–PCL samples, the SEM analysis was performed (Figure 4). SEM micrographs of TPS and TPS–PCL revealed that the neat samples were non-porous (Figure 4), and bulbs of starch are observed in their morphology. A similar presence of spherical shapes was reported for TPS–PLA blends [39]. Although micrographs of the neat samples were recorded at higher magnifications (data not shown), pores were not observed. On the other hand, in the case of samples impregnated with thymol, the presence of pores was recorded, indicating that the proposed scCO_2_ process affects polymer morphology. Additionally, it can be seen that an increase in temperature from 40 to 70 °C led to a decrease in the number of pores. A possible explanation for this behavior is the fact that the gelatinization temperature of corn starch is around 70 °C, the same as the temperature of the scCO_2_ treatment [40]. Additionally, after comparing the samples TPS/T/70/2 and TPS/T/70/6, a lower number of pores is noticed in the first sample, demonstrating that the scCO_2_ process time had an impact on the formation of pores. A micrograph of the TPS–PCL neat sample, in addition to bulbs of starch, shows filaments of PCL (worm-like structures) due to the polymers’ immiscibility. Like in the case of TPS samples, some pores were observed after the impregnation of TPS–PCL with thymol. The absence of visible thymol crystals in all impregnated samples indicates that thymol was incorporated into the polymer matrix during the proposed scCO_2_ process.

Mercury intrusion porosimetry is widely used as a method for determining the pore size distribution of the open pores and/or voids of non-compressible materials. The results of the mercury intrusion porosimetry analysis for the selected samples are presented in Table 2.

The obtained results confirm the findings of the SEM analysis, which indicated that the impregnated samples had a small number of pores, i.e., small porosity. The values of total pore volume for the TPS/T samples are similar, while these values for TPS–PCL/T samples significantly differ. Although an increase in time and temperature (i.e., thymol loading) for TPS–PCL/T samples increased the total pore volume, the values of porosity increased only slightly. According to the obtained results, the sample with the highest thymol loading (TPS–PCL/T/70/6) had the highest values of total pore volume, average pore diameter, and porosity. The results indicate that the addition of PCL to TPS had a significant effect on the morphology of the blend, and provided the possibility to tune the porous morphology.

#### 3.2.3. Thermal Properties of TPS-Based Materials

The TGA analyses were performed to study the influence of the tested scCO_2_ process and thymol loading on the thermal stability of the TPS and TPS–PCL samples. The TG and DTG curves are shown in Figure 5, while Table 3 reports the onset degradation temperature (*T*_onset_), the maximum decomposition temperature (*T*_peak_), the weight loss, and the mass residue of the samples. The TPS samples, neat and impregnated, exhibit a very similar thermal profile with two degradation steps (Figure 5a). The first step is associated with the evaporation of free/bound water and glycerol decomposition from the TPS samples between 25 and 150 °C [41], which represented around 10% of weight reduction. After that, all the samples exhibited a thermal stability until around 280 °C (weight retention was maintained at 80% or higher). The second degradation step of the TPS samples starts around 290 °C. At this point, the weight loss is rapid until reaching the maximum degradation rate of around 319 °C. In this step, the starch chains began to degrade mainly due to the dehydration of the hydroxyl groups and the subsequent formation of unsaturated and aliphatic low molecular weight carbon species [42]. Finally, the last stage of thermal degradation is generally carbonization [43]. Alongside thermograms, DTG curves are given to emphasize the temperatures at which the mentioned changes happened (Figure 5b). Still, it can be seen that the scCO_2_ impregnation with thymol did not have any significant effect on the thermal stability of the examined thermoplastic starch.

On the other hand, the thermograms of the TPS–PCL samples exhibited one additional step (Figure 5c). Like in the case of TPS, the first two steps are related to the evaporation of water (25 to 150 °C) and the degradation of starch (290 to 320 °C). The third step is ascribed to PCL degradation at about 380 °C (polymer chain cleavage via cis-elimination and unzipping depolymerization from the hydroxyl end of the polymer chain) [44]. The DTG curves presented in Figure 5d show all these stages of thermal degradation of each component of the tested materials [41]. The TPS–PCL samples treated with scCO_2_ and impregnated with thymol show a slight decrease in thermal stability compared to the neat sample. It was found that thymol showed a one-step thermal degradation pattern (data not shown) with an onset temperature of 100 °C and complete weight loss at a temperature of 136 °C. Hence, faster initial weight loss could be assigned to the combined effect of evaporation of water and melting and degradation of thymol [45].

According to Table 3, the initial weight loss of the TPS samples was very similar. However, the initial weight loss of the TPS–PCL samples increased from 7% for the neat sample to 15% for the impregnated sample, probably due to the higher concentration and, therefore, degradation of thymol. Treatment with scCO_2_+thymol did not affect the *T*_onset_ and *T*_peak_ of the TPS samples. A slight decrease in *T*_peak_ of starch in the TPS–PCL samples was observed, maybe due to the treatment with scCO_2_. For all samples, the *T*_onset_ of starch is much higher than 100 °C. However, since the processing temperatures of TPS and TPS–PCL did not reach temperatures as high as their *T*_onset_, it can be concluded that the samples show satisfying thermal stability. Similarly, Lerma-Canto et al. [39] reported that the addition of hemp oil to the TPS–PLA blend had a weak effect on the maximum degradation temperature.

Differential scanning calorimetry (DSC) was used to identify the transition temperatures of the untreated samples and the samples treated with scCO_2_+thymol. Table 3 shows the thermal properties of the TPS and TPS–PCL blends in the second heating scan (the thermal history of the materials is erased by the first thermal scanning on DSC, not reported). In Table 4, the *T*_g_ of both tested polymers is reported. As it can be seen, the *T*_g_ of the TPS occurs at a very high temperature due to the removal of water (TPS plasticizer) during the isothermal step of DSC analysis (60 min at 110 °C). Indeed, water is able to increase the free volume of the polymer since it breaks intra- and intermolecular hydrogen bonding by promoting macromolecular mobility. By eliminating the water and starch, like all polysaccharide, it induces the development of more rigid and tight conformation resulting in very high *T*_g_ [46]. Similarly, Liu et al. [47] studied the corn starch films with different moisture contents and observed that the *T*_g_ increased with a decrease in moisture content in the starch sample. The *T*_g_ of starch in neat TPS and TPS–PCL samples is 149 and 152 °C, respectively. Then, once again, treatment with scCO_2_+thymol of TPS and TPS–PCL generally caused the shift of *T*_g_ to higher values, indicating that scCO_2_+thymol also affects the conformation of starch. Similarly, it was reported that the addition of *Cymbopogan citratus* fiber to TPC increases the *T*_g_ (from 81.5 to 137.7 °C with an increase in fiber content from 0 to 60 wt%) [32]. Contrary to these results, Lerma-Canto et al. [39] reported that the addition of hemp oil to the TPS–PLA blend decreases its *T*_g_ and concluded that the additive is acting as a lubricant, decreasing interaction between polymer chains.

The *T*_g_ of PCL (*T*_g_,_PCL_) in a neat TPS–PCL blend was detected at −74 °C. Furthermore, it is found that *T*_g_,_PCL_ is not influenced by thermal treatment and it does not undergo significate change after being exposed to scCO_2_+thymol. In any case, the presence of two *T*_g_ (for starch and PCL) suggested scarce polymer miscibility, as expected by the hydrophilic and hydrophobic nature of the two polymers in the blend.

Conversely, an interesting change was observed regarding the melting of the PCL upon treatment with scCO_2_+thymol. In the neat TPS–PCL and TPS/T/70/2 samples, a characteristic endothermic peak was detected in the temperature range of 45–60 °C attributed to the melting of the PCL [48,49]. According to Table 4, the PCL melting peak became less intense with the influence of the scCO_2_+thymol treatment (the value of the melting enthalpy, Δ*H*_m_, slightly decreased). Additionally, with a further prolongation of the scCO_2_+thymol treatment to 6 h, this endothermic peak of PCL completely disappears, indicating that the processing method had a significant effect on PCL structure and could help to tailor properties for desired needs. Some authors used the DSC method to confirm the obtained amorphous PCL [50,51]. Guo et al. [50] performed in situ copolymerization to prepare a poly(diol citrate)/poly(caprolactone) composite elastomer to produce a full-amorphous structure. Likewise, Thomas and Nair [51] have prepared degradable polyesters by a polycondensation technique with citric acid and polycaprolactone triol and showed that the material was amorphous due to the absence of a crystalline melting peak.

### 3.3. Thymol Release from TPS-Based Materials

Food packaging materials with a high loading of active agents and a highly responsive nature have been in high demand in recent decades. As already mentioned, thymol is a phenolic monoterpene that has received considerable attention as an antimicrobial agent [22,27]. It also can be used as a possible food antioxidant [23]. Due to that fact, it was important to investigate in vitro thymol release from TPS and TPS–PCL samples at the storage temperature of the food on cooling shelves and room temperature in water (6 °C and 25 °C). Figure 6a shows plots of the concentration of released thymol from TPS and TPS–PCL samples treated with scCO_2_+thymol at 70 °C for 6 h versus different times. It can be seen that thymol release from the TPS/T/70/6 sample reached equilibrium within 6 h independent of the release temperature (Figure 6a, insert). The maximum concentration of thymol achieved for the release period was ca. 15 mg_th_/g_s_. Additionally, it can be seen that thymol release from the TPS–PCL blend was much higher compared to the TPS sample. In the case of TPS–PCL samples, the final concentration of released thymol, regardless of the release temperature, was very similar (ca. 73 mg_th_/g_s_). This result was expected, considering that thymol loading was significantly higher in the TPS–PCL sample than in the TPS sample (*L*_TPS/T/70/6_ = 2.1% and *L*_TPS-PCL/T/70/6_ = 15%). It is evident that a longer time was required to reach equilibrium when thymol was released from the TPS–PCL sample (Figure 6a). Moreover, an initially slightly faster thymol release rate was observed at 6 °C in the first 6 h of the release test, but as the experiment progressed, the release was faster at 25 °C. Overall, the controlled release of thymol from the TPS–PCL/T/70/6 sample was achieved over the course of 7 days, which is a promising result considering that after further processing this material could possibly be applied as a smart food packaging material.

The release kinetics of thymol at 6 °C from the TPS and TPS–PCL samples impregnated at different conditions are presented in Figure 6b,c. It can be seen that time and temperature of scCO_2_+thymol treatment played a significant role in the release profile. In the case of TPS samples, the amount of released thymol decreased in the following order: TPS/T/70/2 (*C* = 34.4 mg_th_/g_s_) > TPS/T/40/6 (*C* = 22.0 mg_th_/g_s_) > TPS/T/70/6 (*C* = 15.3 mg_th_/g_s_) with the increase in thymol loading. Davoodi et al. [52] showed that the decrease in the thymol release might be attributed to the encapsulation of thymol in the starch chain, which could indicate that a stronger interaction of starch and thymol formed as the scCO_2_+thymol treatment progressed. Hence, the lower release was obtained from the samples exposed to scCO_2_+thymol for a longer time. Additionally, the samples TPS/T/40/6 and TPS/T/70/6 reached equilibrium much faster compared to TPS/T/70/2. The first two samples reached equilibrium within the first 6 h of release, while 2 days were required for TPS/T/70/2.

As can be seen in Figure 6c, the release rate of thymol was much higher in all TPS–PCL samples compared to the TPS samples. The highest release was obtained from TPS– PCL/T/40/6, with the final concentration reaching 87.2 mg_th_/g_s_. The samples TPS–PCL/T/70/2 and TPS–PCL/T/70/6 exhibited similar release kinetics to thymol. Both samples released thymol for 7 days (the concentration of thymol was ca. 73 mg_th_/g_s_) in a controlled manner. These findings reveal that the addition of 20 wt.% PCL to TPS had an important role in the thymol loading and directly on the release of thymol. Not only is PCL a biodegradable polymer, it is also widely used in systems for the controlled release of substances, recommending it for the development of functional materials [53].

## 4. Conclusions

This study showed the applicability of the SSI process and of scCO_2_ as a green medium for the incorporation of thymol into thermoplastic starch (TPS) and thermoplastic starch blend with poly (*ε*-caprolactone) (TPS–PCL) for the first time. The results of the scCO_2_-assisted process show that thymol loading was strongly influenced by the process conditions (temperature and operating time) as well as the presence of PCL in TPS. The methods of characterization confirmed the presence of thymol in the polymer matrix and revealed the effects of thymol and PCL on final material properties. While the presence of PCL did not significantly affect the thermal properties of TPS, it significantly contributed to the increase in thymol loading and subsequently the controlled thymol release. The controlled release of thymol from TPS was achieved for 6 h while TPS–PCL released thymol for 7 days. Therefore, it is possible to tailor the release profiles of bioactive components to specific needs, which is valuable for the food industry. The results confirm the indication that TPS and TPS–PCL are promising materials, which could be impregnated with a bioactive component in an environmentally friendly way. This comprehensive study provided valuable data on the interactions of the ternary system scCO_2_-polymer-bioactive component and provided guidance for the further development of starch-based biodegradable active materials that could be used for food packaging.

## Figures and Tables

**Figure 1 polymers-14-04360-f001:**
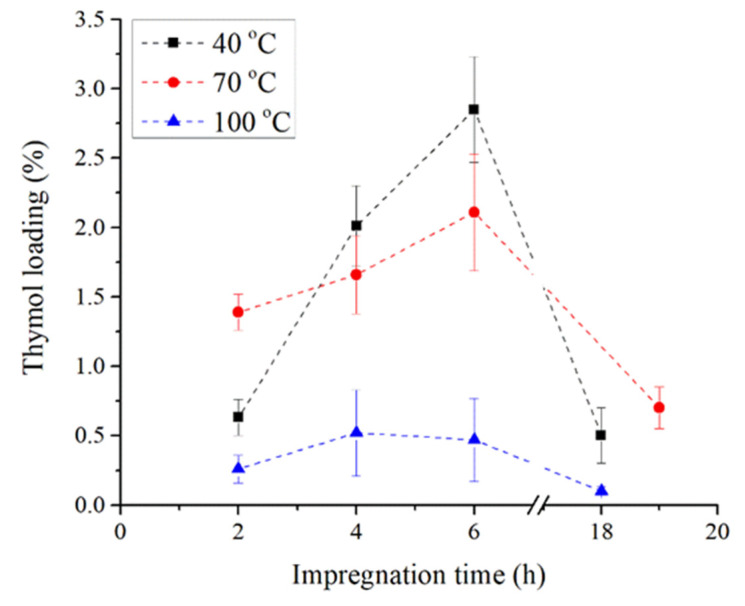
Kinetic of thymol loading into TPS.

**Figure 2 polymers-14-04360-f002:**
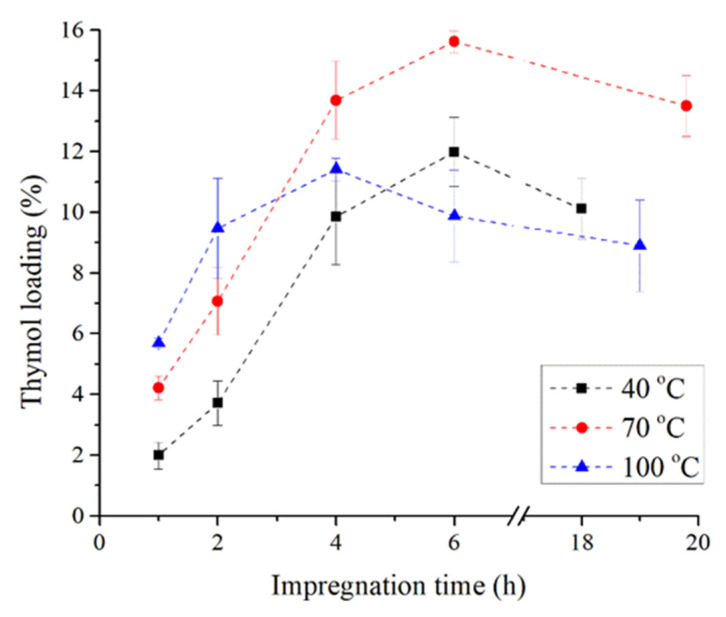
Kinetic of thymol loading into TPS–PCL.

**Figure 3 polymers-14-04360-f003:**
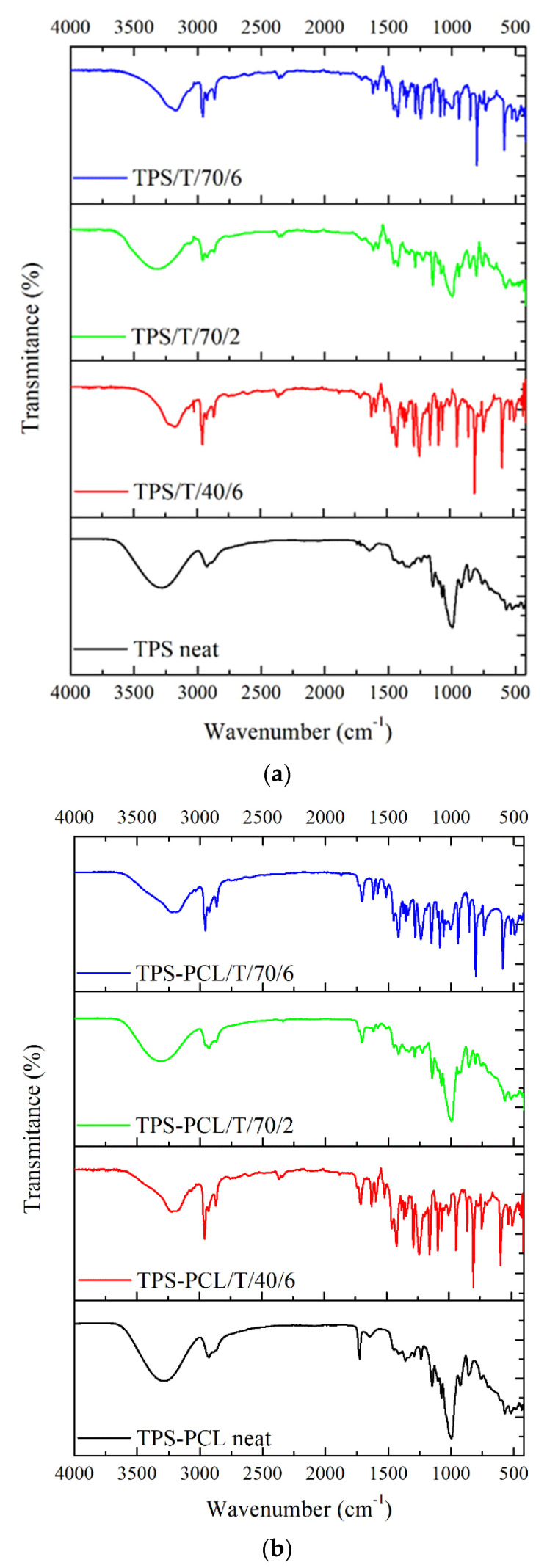
FTIR spectra of neat and thymol-impregnated: (**a**) TPS samples; (**b**) TPS–PCL samples.

**Figure 4 polymers-14-04360-f004:**
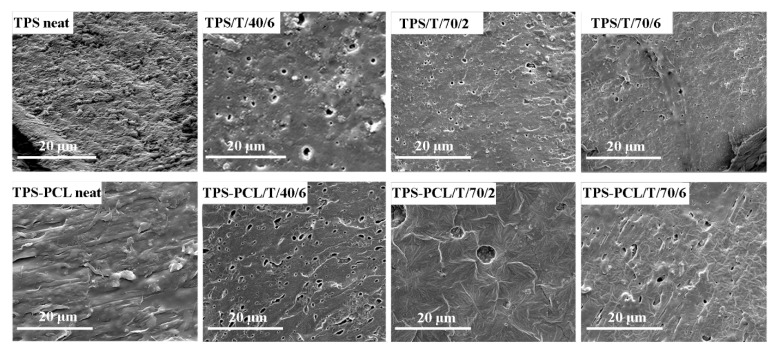
SEM micrographs of the fractured cross-section of neat and impregnated TPS and TPS–PCL samples.

**Figure 5 polymers-14-04360-f005:**
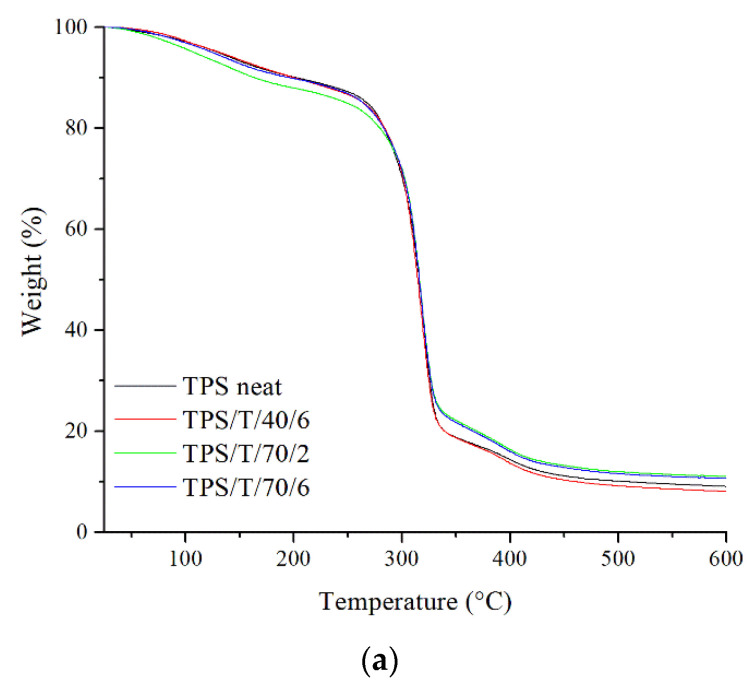
The TGA (**a**,**c**) and DTG (**b**,**d**) curves of neat and impregnated TPS and TPS–PCL samples.

**Figure 6 polymers-14-04360-f006:**
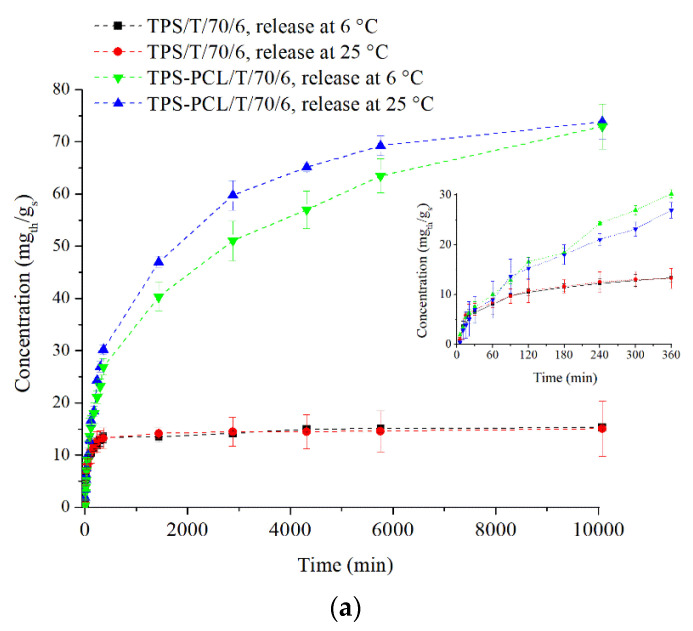
(**a**) Thymol release at 25 °C and 6 °C from TPS/T/70/6 and TPS–PCL/T/70/6 (inserted graph shows the release profile during the first 6 h), (**b**) cumulative release of thymol at 6 °C from TPS samples, and (**c**) cumulative release of thymol at 6 °C from TPS–PCL samples.

**Table 1 polymers-14-04360-t001:** List of samples impregnated with thymol (T).

Polymer	Temperature, °C	CO_2_ Density, kg/m^3^	Time, h	Sample Abbreviation
TPS	40	910	2	TPS/T/40/2
			4	TPS/T/40/4
			6	TPS/T/40/6
			18	TPS/T/40/18
	70	788	2	TPS/T/70/2
			4	TPS/T/70/4
			6	TPS/T/70/6
			19	TPS/T/70/19
	100	662	2	TPS/T/100/2
			4	TPS/T/100/4
			6	TPS/T/100/6
			18	TPS/T/100/18
TPS–PCL	40	910	1	TPS-PCL/T/40/1
			2	TPS-PCL/T/40/2
			4	TPS-PCL/T/40/4
			6	TPS-PCL/T/40/6
			18	TPS-PCL/T/40/18
	70	788	1	TPS-PCL/T/70/1
			2	TPS-PCL/T/70/2
			4	TPS-PCL/T/70/4
			6	TPS-PCL/T/70/6
			20	TPS-PCL/T/70/20
	100	662	1	TPS-PCL/T/100/1
			2	TPS-PCL/T/100/2
			4	TPS-PCL/T/100/4
			6	TPS-PCL/T/100/6
			19	TPS-PCL/T/100/19

**Table 2 polymers-14-04360-t002:** Textural characteristics of the prepared TPS-based samples.

Sample	*V* _p_	*S* _s_	*D* _p,average_	*P*
	(mm^3^/g)	(m^2^/g)	(nm)	(vol%)
TPS/T/40/6	8.5	0.9	8	0.4
TPS/T/70/2	12.2	3.1	8	1.9
TPS/T/70/6	12.9	0.5	74	1.7
TPS-PCL/T/40/6	15.0	1.9	40	2.0
TPS-PCL/T/70/2	25.7	4.3	36	3.2
TPS-PCL/T/70/6	40.6	2.1	148	5.6

*V*_p_—total pore volume; *S*_s_—specific surface area; *D*_p,average_—pore diameter average; *P*—porosity.

**Table 3 polymers-14-04360-t003:** Thermal properties of neat and impregnated TPS and TPS–PCL samples.

Sample	*W*_L_ (%) (25-150 °C)	Starch		PCL		*M*_R_ (%) at 600 °C	2nd Heating	
		*T*_onset_ (°C)	*T*_peak_ (°C)	*T*_onset_ (°C)	*T*_peak_ (°C)		*T*_g TPS_ (°C)	*T*_g PCL_ (°C)
TPS neat	6.6	293.2	319.7			9.0	149	/
TPS/T/40/6	6.2	294.2	318.3			8.1	160	
TPS/T/70/2	8.8	296.6	319.2			10.4	162	
TPS/T/70/6	7.1	294.7	318.2			10.6	163	
TPS–PCL neat	7.4	296.9	319.0	382.3	401.6	8.4	152	−74
TPS–PCL/T/40/6	12.5	300.6	316.6	381.9	400.9	9.8	154	−73
TPS–PCL/T/70/2	9.9	300.8	316.6	383.6	404.3	9.1	156	−72
TPS–PCL/T/70/6	15.5	297.2	318.0	378.4	402.4	8.1	160	−73

*W*_L_—weight loss; *M*_R_—mass residue.

**Table 4 polymers-14-04360-t004:** Thermal parameters of TPS and TPS–PCL blends from DSC.

Sample	2nd Heating Scan			
	*T*_g TPS_ (°C)	*T*_g PCL_ (°C)	Δ*H*_m_ (J/g)	*T*_m PCL_ (°C)
TPS neat	149	/	/	/
TPS/T/40/6	160			
TPS/T/70/2	162			
TPS/T/70/6	163			
TPS–PCL neat	152	−74	2.017	54
TPS–PCL/T/40/6	154	−73	no peak	
TPS–PCL/T/70/2	156	−72	1.583	55
TPS–PCL/T/70/6	160	−73	no peak	

## Data Availability

The data presented in this study are available upon request from the corresponding author.

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
