# Peer review of "Supercritical CO2 Impregnation of Thymol in Thermoplastic Starch-Based Blends: Chemico-Physical Properties and Release Kinetics"

_polymers, 2022, doi:10.3390/polym14204360_

Round 1

Reviewer 1 Report

The supercritical CO2 impregnation process for thymol loading into TPS and TPS-PCL has been studied, more data are needed to support some results and more insight is also needed:

Comment 1: On Page 6 Line 203-205 “An increase in the impregnation time to 20 h led to a decrease in thymol loading into TPS with the final values of 0.5, 0.7, and 0.1% at temperatures of 40, 70, and 100 °C, respectively.”. The authors mentioned that the content of thymol in TPS decreased when the impregnation time reached 20h. However, figure1 has no relevant data to support the authors’ statement. Please give a reasonable explanation.

Comment 2: The authors mentioned in the manuscript that the increase of thymol content in TPS-PCL was attributed to the high affinity between thymol and PCL. However, the content of thymol in the blends is not only related to the affinity of PCL, but also to the compatibility of TPS and PCL. If the compatibility between TPS and PCL is poor, the voids between the phase interfaces will also store thymol. The authors are requested to carefully consider the reasons for the increase in thymol content and to provide relevant evidence for further clarification.

Comment 3: The same problem exists in Figure 2. It is mentioned that the thymol content decreases after the impregnation time reaches 20 h, but no relevant data are provided in Figure 2.

Comment 4: On Page 15 Line 459-461 “They found that an increase of PCL content increased thymol loading due to the plasticizing effect of the PCL on the polymer matrix and increased solubility of the substances in it”. What is the mechanism of the plasticizing effect of PCL on polymer matrix? How does one polymer plasticize another polymer? Please provide additional relevant experiments.

Comment 5: All figures in the manuscript require further improvement.

Author Response

# Reviewer 1

The supercritical CO2 impregnation process for thymol loading into TPS and TPS-PCL has been studied, more data are needed to support some results and more insight is also needed:

Response: Additional data is now included into to the Manuscript. Also, new and/or improved SEM images are added.

Comment 1: On Page 6 Line 203-205 “An increase in the impregnation time to 20 h led to a decrease in thymol loading into TPS with the final values of 0.5, 0.7, and 0.1% at temperatures of 40, 70, and 100 °C, respectively.”. The authors mentioned that the content of thymol in TPS decreased when the impregnation time reached 20h. However, figure1 has no relevant data to support the authors’ statement. Please give a reasonable explanation.

Response: The Figure 1 is corrected. It now contains data reported in lines 203-205.

Comment 2: The authors mentioned in the manuscript that the increase of thymol content in TPS-PCL was attributed to the high affinity between thymol and PCL. However, the content of thymol in the blends is not only related to the affinity of PCL, but also to the compatibility of TPS and PCL. If the compatibility between TPS and PCL is poor, the voids between the phase interfaces will also store thymol. The authors are requested to carefully consider the reasons for the increase in thymol content and to provide relevant evidence for further clarification.

Response: Thank you for the comment. Our previous study showed that chemical modification of PCL prior to blending with TPS increases compatibility between TPS and PCL. The information about chemical modification of PCL was mentioned in lines 107-108. The comment about thymol impregnation results is now improved in following manner “One of the reasons for the increase in thymol loading with an increase in PCL content in TPS from 0 to 20% could be the high affinity of thymol shown towards PCL during scCO2 impregnation processes. Namely, it was previously reported that thymol loading is increasing with an increase of the PCL content in PLA [29] and that thymol loading into neat PCL of 14.1% can be achieved after only 2 h at 30 MPa and 40 °C [25,30]. Under the same conditions, thymol loadings into TPS and TPS-PCL obtained in this study were 0.6 and 3.7%, respectively. Additional reason for the increase in thymol loadings with PCL content in TPS could be possible existence of voids between the phase interfaces due to low compatibility between TPS and PCL polymers. However, assessing the influence of the mentioned factors requires additional comprehensive investigation”.

Comment 3: The same problem exists in Figure 2. It is mentioned that the thymol content decreases after the impregnation time reaches 20 h, but no relevant data are provided in Figure 2.

Response: The Figure 2 is corrected. It now contains data reported in lines 223-225.

Comment 4: On Page 15 Line 459-461 “They found that an increase of PCL content increased thymol loading due to the plasticizing effect of the PCL on the polymer matrix and increased solubility of the substances in it”. What is the mechanism of the plasticizing effect of PCL on polymer matrix? How does one polymer plasticize another polymer? Please provide additional relevant experiments.

Response: Thank you for the comment. Authors regret this mistake. The correct statement would be “Supercritical CO2 has a plasticizing effect on PCL, which favors sorption of thymol”. However, upon review, authors decided to remove last three statements from the Manuscript.

Comment 5: All figures in the manuscript require further improvement.

Response: All figures are now corrected.

Reviewer 2 Report

Authors state they are developing biodegradable bioactive packagings from biobased polymers and blends. However, in the Abstract beginning, the packagings are stated to be able to reduce waste. How this will be possible? Why is not degradability studied? What about sample shaping into packaging? What kind of packaging?

Additionally, the results and discussion are closer to the paper title than to the abstract and introduction sections. These must be written correctly prior publishing.

With this, I can only recommend minor revisions.

Author Response

# Reviewer 2

Authors state they are developing biodegradable bioactive packagings from biobased polymers and blends. However, in the Abstract beginning, the packagings are stated to be able to reduce waste. How this will be possible? Why is not degradability studied? What about sample shaping into packaging? What kind of packaging?

Additionally, the results and discussion are closer to the paper title than to the abstract and introduction sections. These must be written correctly prior publishing.

With this, I can only recommend minor revisions.

Response: Thank you for the comment. Authors regret the ambiguity of statements. Abstract, Introduction, and Conclusion are now rephrased.

The aim of the present study was to investigate starch-based materials, prepared in an environmentally friendly way and from renewable resources, which are suitable for development of biodegradable active food packaging. For the first time an environmentally friendly technique of supercritical CO2 impregnation was employed for processing of TPS-based polymers. A comprehensive study on the effect of supercritical CO2 processing conditions on TPS-based materials properties was performed and explained in detail. This comprehensive study provided valuable data on the interactions of ternary system scCO2-polymer-bioactive component and gave guidance for the further development of biodegradable active food packaging. Namely, our first study on TPS and TPS-PCL materials preparation (Orteg-Toro et al. 2016) reported that developed starched-based materials showed gas and water vapor barrier properties comparable to some synthetic plastics commonly used for food packaging (this information is now added to the Manuscript). Current study investigated potential for functionalization of TPS-based materials with a bioactive component in an environmentally friendly manner. Our next study will investigate biodegradability of obtained materials, migration in different food simulants, and the stability of TPS-based material as rigid food packaging and packed food over the storage time.

Additionally, it was stated that TPS can be transformed by injection or blow molding into the final products suitable for food packaging in lines 42-45. Also, the data on biodegradability of TPS-PCL materials is now included in next lines.
